# Prevalence, Screening Practices and Risk Stratification for Diabetic Foot Complications in Primary Healthcare Clinics: A Cross-Sectional Study in Gauteng Province, South Africa

**DOI:** 10.3390/ijerph22121794

**Published:** 2025-11-27

**Authors:** Simiso Ntuli

**Affiliations:** Department of Podiatry, Faculty of Health Sciences, University of Johannesburg, Johannesburg 2006, South Africa; sntuli@uj.ac.za

**Keywords:** diabetic foot, diabetic foot complications, diabetic foot screening, primary healthcare, risk factors, screening practices, chronic disease management, diabetes mellitus, foot ulcers, preventive care

## Abstract

**Background/Objective**: The incidence of diabetic foot complications has been increasing in South Africa. While foot screening at the primary healthcare level is crucial for preventing these complications, the growing number of cases in hospitals indicates that persons with diabetes are not undergoing routine screening in primary healthcare settings. **Method**: This cross-sectional descriptive observational study was conducted at five community healthcare centres, one in each municipality in Gauteng. Participants included persons with diabetes who presented at these facilities for their routine diabetes care. Data were collected, which included patient demographics, a history of screening, risk factors for diabetic foot, and an assessment of diabetic foot was conducted for each participant. **Results**: A total of 597 diabetic patients volunteered for this study. Only 10% (n = 60) had received a diabetic foot assessment. No patient had been risk-stratified; the results showed that 30% (n = 178) were very low risk and 17% (101) were at high risk. Active ulcers were recorded in 19% (116), and 18% (106) were in remission. Neuropathy was recorded in 33% (197), peripheral arterial disease in 22% (131), and a history of amputation was recorded in 17% (103). **Conclusions**: Implementing routine diabetic foot assessment and risk stratification at the PHC level could be key in preventing diabetic-related complications.

## 1. Introduction

The International Diabetes Federation (IDF) estimates that approximately 4.2 million individuals are currently living with diabetes in South Africa (RSA) [1,2]. Diabetes and its complications are strongly associated with modifiable risk factors and determinants, which should be a significant public health concern in RSA. National guidelines for the management of diabetes recommend that patients be reviewed by a healthcare professional, such as a nurse or doctor, at least four times annually [3]. In practice, most patients attend monthly clinic visits to monitor random blood glucose levels, blood pressure, and weight and collect prescribed medications [3,4]. Studies have found that most South Africans with T2D are managed at the PHC level, where the standard of care is inadequate; only 10–30% of patients in the public health system achieve glycaemic control or an HbA1c of <7.0% [5,6]. Despite strong evidence supporting intensive blood glucose control, many patients with type 2 diabetes remain on suboptimal therapy. Numerous studies have linked this gap to healthcare professionals not intensifying treatment when clinically indicated, contributing to poor glycaemic outcomes [6].

In Gauteng province alone, where this study was conducted, an estimated 1.9 million or 12.0% of the population are living with type 2 diabetes [7], and their management is undertaken primarily through 368 primary healthcare (PHC) facilities distributed across the region [8]. Persons with diabetes are more at risk of developing foot problems, with those affected experiencing higher rates of foot ulceration, lower-limb amputation and premature death [9,10].

Evidence suggests that between 25% and 34% of individuals with diabetes are at risk of developing a diabetic foot ulcer (DFU) during their lifetime [11,12]. These ulcers are primarily preceded by risk factors, including peripheral neuropathy, ischaemia, and foot deformities, and are the leading cause of lower-extremity amputations. Approximately 85% of lower extremity amputations in individuals with diabetes are preceded by foot ulcers [13,14]. Early interventions focused on identifying and preventing the onset or progression of DFUs can reduce the incidence of these ulcers by up to 50% [15,16] and decrease the likelihood of amputations by approximately 50–85% [17,18]. This requires regularly scheduled foot screening to identify individuals at risk of developing DFUs effectively [19].

In RSA, 80% of persons with diabetes are managed at the PHC level rather than in hospital settings [16,17]. This is where prevention efforts can be both practical and scalable. Primary healthcare facilities serve as the patients’ closest and first point of contact, and they could play a vital role in preventing and managing diabetic foot ulcer (DFU) risk factors. Although PHC plays a crucial role in diabetes management, significant gaps exist in the quality of care at this level [4,6]. Diabetic foot screening at this level of care is a critical concern [20,21,22,23]. Despite local and international recommendations [3,24], foot screening has been inconsistently implemented in PHC facilities in South Africa [25]. Implementing routine screening for all persons with diabetes at the PHC level can enhance the early detection of complications such as infection, ulceration, and ischemia. Identifying high-risk patients promptly and referring them to multidisciplinary teams is crucial in preventing diabetic foot ulcers and their severe consequences.

International guidelines recommend annual foot assessments for all persons with diabetes, with more frequent evaluations for those with prior ulceration, neuropathy, or foot deformities [19]. Patients with non-healing DFUs should be referred to higher levels of care within two weeks [26,27]. Within the RSA context, delays in recognition and referral remain common, often due to gaps in screening, clinical training and the absence of structured referral pathways [28]. This leads to a growing burden of serious diabetes complications, which significantly contribute to morbidity, loss of function, and increased mortality [29,30]. In Gauteng province, South Africa, up to 60% of diabetes-related hospitalisations in the province are attributed to foot complications arising from DFUs [31,32,33]. Literature suggests that 20% of people who develop a DFU will require lower-extremity amputation, either minor (below the ankle), major (above the ankle), or both [34], and 10% will die within 1 year of their first DFU diagnosis [35,36].

This study addresses a significant knowledge gap in the South African diabetic foot care literature. To date, clinical data on diabetic foot complications have primarily been obtained from tertiary-level facilities. Conversely, research at the primary healthcare level has largely relied on retrospective patient file reviews, which are useful for identifying documentation gaps and inconsistent screening practices but offer limited insight into the actual diabetic foot clinical presentation and risk profiles of patients, particularly those presenting with early signs of diabetic foot complications. Notably, no prior studies in South Africa have produced empirical data on the clinical presenting risk factors or the stratification of patients by risk category at the primary healthcare level, representing a substantial shortfall in the evidence base. Presenting diabetic foot risk factors and risk stratification are fundamental to effective diabetic foot prevention, directly informing immediate clinical decisions, including referral urgency, follow-up intensity, and resource allocation. In the absence of such data, policymakers and healthcare providers lack the necessary information to tailor interventions to individual patient risk levels, potentially delaying care and increasing the likelihood of adverse outcomes.

This study generated real-time data on both the presence and nature of diabetic foot risk factors and the stratification of patients into risk categories through direct clinical assessments. The findings confirm the inadequacy of current screening practices and provide a foundation for more responsive, risk-informed care. The study’s contribution is both timely and novel, offering actionable insights that can inform public health policy, strengthen primary healthcare protocols, and reduce the burden of diabetic foot complications across all levels of care.

## 2. Materials and Methods

### 2.1. Research Area

This cross-sectional descriptive observational study was conducted at five community healthcare centres (CHC) in Gauteng, selected using single-stage cluster sampling. Gauteng is the smallest province of South Africa, covering an area of 18,178 km^2^, approximately 1.4% of the country’s total surface area. Despite its small size, Gauteng is the most populous province, with a population of 15.83 million, accounting for 26.3% of the national population. Gauteng has 368 PHC facilities that provide health services to approximately 85% of its 15.83 million residents, including 38 CHC facilities [7]. The prevalence of type 2 diabetes within Gauteng province is estimated to be between 11.1% and 12.8% [37].

### 2.2. Study Population

The study population included all patients with type 2 diabetes in Gauteng, and the target population included persons with diabetes presenting at the participating clinics. Eligible participants were those aged > 18 years, diagnosed with diabetes mellitus, and capable of providing informed consent. This study was approved by the University of Johannesburg Research Ethics Committee (REC-1934-2023). Informed consent was obtained from all study participants, per the guidelines outlined in the revised Declaration of Helsinki [38].

### 2.3. Research Participants

A purposive convenience sampling technique was used to invite consecutive persons with diabetes attending the selected clinics to participate in the study. The sample size calculation was based on a single population proportion formula using a 95% confidence level, a margin of error of 5%, and a 50% frequency. This resulted in an estimated sample size of 385. To account for potential contingencies, such as refusal and incomplete or missing data, an additional 25% was added to the sample size. This resulted in an estimated sample size of 480 patients in this study.

The researcher used a convenience sampling method to recruit participants from the community clinics. This approach was chosen because operational constraints in these settings made random sampling impractical without disrupting the routine clinical workflows and patient care. Convenience sampling allowed the researcher to include participants who were easily accessible during the study period, enabling timely data collection despite the project’s resource and time constraints. While this method facilitated practical implementation, the researcher recognises that it could have introduced selection bias and limited sample representativeness.

### 2.4. Study Questionnaire

The data collection form comprised three sections. The first section covered the participants’ demographic parameters. The second section focused on presenting diabetic foot risk factors, including a history of diabetic foot disease. The third focused on risk stratification using the IWGDF Risk Stratification System [24]. Participants signed a consent form and were assessed by the researcher in a private consulting room after their routine diabetes consultation. The total time to complete the questionnaire and undertake a diabetic foot assessment was 20 min (Appendix A). The questionnaire was adapted from a questionnaire developed for a Master’s project that examined the need for podiatrists as part of the primary healthcare team [39].

The questionnaire used in this study was adapted from an instrument that had been validated during a previous master’s research project. The validation process included an expert review to establish content validity, participant feedback for face validity, and reliability testing using Cronbach’s alpha (α = 0.82), indicating good internal consistency. The adapted questionnaire was piloted with ten participants for the current study to assess its clarity, relevance, and feasibility in the intended context. Feedback from the pilot was used to make minor refinements to ensure that the tool was appropriate for the main study. The pilot study participants were not part of the large study.

### 2.5. Data Collection Procedure

Data were collected from adult participants attending scheduled diabetes clinic appointments at selected PHC facilities. A multi-method approach was employed, comprising a researcher-administered questionnaire, retrospective patient file review, and foot screening. The questionnaire captured demographic information, presented diabetic foot risk factors, and explored the history of diabetic foot assessments conducted at the respective PHC clinics. The researcher administered the questionnaire during the clinic visit to ensure consistency and minimise bias.

Foot screening was conducted using standard clinical procedures (skin integrity, foot deformities, palpation of peripheral pulsations, and peripheral neuropathy). No specialised equipment was used except for the assessment of peripheral neuropathy, which was performed using a 5.07/10 g Semmes-Weinstein monofilament (SWM). Sensory testing was conducted at five anatomical sites on each foot: the pulp of the first and third toes and the first, third, and fifth metatarsophalangeal joints (MPJs), totalling ten sites. Each site was tested twice in a blinded and arrhythmic manner to reduce the response bias. Areas with calluses, ulcers, or scars were excluded from testing. Sensation was classified as normal if eight or more sites were detected, and abnormal if seven or fewer sites were detected.

Additionally, a retrospective review of each consenting participant’s medical file was conducted to extract relevant clinical history and verify the documentation of prior diabetic foot assessments. This triangulated approach allowed for a comprehensive evaluation of current screening practices and historical care patterns.

### 2.6. Data Analysis

The collected data were captured in an Excel spreadsheet and coded before being loaded into a statistical computer package (SPSS version 29) NC. Means (±SD) were used to summarise continuous variables, and percentages were used to summarise categorical variables. The Chi-square test was used to test for associations between categorical variables. A *p*-value of less than 0.05 was considered statistically significant.

## 3. Results

This study included 597 participants, whose baseline characteristics are outlined in Table 1. The study population was predominantly female, with 56% representation, and had a mean age of 47.17 years (standard deviation ± 11.7 years). The age range was 37–73 years. Most participants were African Black (69%), and the majority were female (56%).

### 3.1. Multimorbidity and Comorbidities

Persons with diabetes are known to present with multimorbidity and comorbidities, as shown in Table 2. In this study, 35% of persons with diabetes presented with only diabetes, and hypertension was the most commonly recorded comorbid disorder. The majority of patients, 58% (346/597), had an RBG of 6–10 mmol/L. There was a significant association with the multibobidity and diabetic foot risk factors, such as loss of sensation, current ulceration, absence of pulses (*p* = 0.021) and random blood glucose of more than 11 mmol/L (*p* = 0.003).

### 3.2. Recorded Diabetic Foot Complications or Risk Factors

A number of 33% (198/597) presented with neuropathy, 19% (116/597) had active ulcers, and 15% (89/597) had a history of amputation. The results indicated a significant relationship between current ulceration and loss of protective sensation (*p* = 0.017) and poor circulation (*p* = 0.037). Additionally, an association was found between the ulcer site and foot deformity, particularly prominent metatarsals (*p* = 0.002). The full findings of the recorded diabetic foot complications and risk factors are presented in Table 3.

### 3.3. Diabetic Screening

In this study, 10% (60/597) of persons with diabetes had undergone diabetic foot screening in the previous 12 months. The overall findings are presented in Table 4.

### 3.4. Diabetic Foot Risk Stratification

Risk stratification of the study cohort using the IWGDF classification system revealed that 30% (178 out of 597) of patients were categorised as having a very low risk, and 17% were identified as high risk. Importantly, patients in the high-risk category presented with active foot pathology, indicating a significantly increased likelihood of complications such as ulceration or infection. These findings illustrate the distribution of risk levels within the cohort and provide a basis for targeted clinical interventions aligned with individual patient risk profiles. The overall findings are presented in Table 5.

## 4. Discussion

The findings of this study underscore the untapped potential of PHC facilities in mitigating diabetic foot complications (DFCs). As the primary point of contact for most persons with diabetes, PHC clinics are strategically positioned to implement routine foot screening and early intervention. However, diabetic foot screening at the PHC level is not optimal. The author suggests that this potential is not being fully realised, possibly due to the lack of structured, institutionalised screening protocols. This disconnect between policy and practice, as highlighted in the background, contributes to delayed recognition of complications and increased risk of adverse outcomes such as ulceration and amputation. Addressing this gap is essential to improving patient outcomes and reducing the burden of diabetic foot disease in South Africa.

This study identified key risk factors associated with diabetic foot complications, with significant correlations observed between poor glycaemic control and current ulceration. Patients with random blood glucose levels of 11–15 mmol/L were more likely to present with foot complications (*p* = 0.003), reinforcing the importance of effective glucose management.

Multimorbidity also showed a significant association, particularly among persons with diabetes and two additional chronic conditions (*p* = 0.021), suggesting that cumulative disease burden increases vulnerability. Structural and neurological factors, such as numbness (LOPS), non-palpable pulses, pes cavus, and prominent metatarsal heads, were also significantly linked to foot complications. These findings highlight the need for integrated screening and management strategies and underscore the urgency for policy-driven research to strengthen diabetic foot care pathways and inform resource allocation.

Screening for individuals at risk of diabetic foot complications (DFCs) involves identifying clinical and social factors directly associated with ulceration. Key risk factors include loss of protective sensation, peripheral arterial disease, foot deformities, calluses, a history of previous ulceration, inability to perform self-care, and inadequate footwear [40]. A quick and reliable screening process is essential for effectively assessing risk factors. Regular screening is necessary to detect future changes. Early identification of risk factors enables healthcare professionals to provide appropriate foot care, offer targeted patient education, perform risk stratification, and make timely referrals for at-risk patients. The link between DFUs and diabetic-related foot and lower extremity amputations (DRFLEA) is clear, as DFU precedes 85–90% of amputations [13,14]. Most diabetic foot complications can be prevented through early screening [16,41]. Foot screening aims to identify individuals at risk of developing foot ulcers. In the RSA context, PHC facilities provide an ideal setting for continuous and regular diabetic foot screening.

Primary healthcare clinics are the first point of contact for 80% of patients with type 2 diabetes in South Africa [6,20]. However, evidence from the current study indicates that diabetic foot screening is often neglected. Screening practices in these clinics remain suboptimal due to many factors that require further investigation. However, it is important to remember that healthcare professionals at PHC facilities could face systemic challenges in delivering quality diabetic foot care. These include limited resources, insufficient training among healthcare workers, and a lack of standardised screening protocols and referral pathways [31,42]. Nurses working at PHC facilities see between 40 and 50 patients a day, which limits their consultation times with each patient [43,44].

Without diabetic foot screening, the early signs of diabetic foot complications are often unnoticed. Important assessments that can identify peripheral neuropathy, peripheral vascular disease, increased plantar pressure, and foot deformities are frequently not performed [17,18].

While early detection of risk factors does not guarantee successful treatment, patients with diabetic foot complications (DFC) must receive appropriate care, education and risk categorisation. Early detection at a lower level of care could significantly influence ongoing efforts to enhance the integration of foot health care within PHC settings. This study showed that patients presenting at this level have a serious need for regular screening. Patients in this study presented with active ulcers (19%), ulcers in remission (18%), and a history of amputation (15%).

Previous ulceration or amputation is the highest risk factor for DFU recurrence or relapse. Notably, only 10% of the persons with diabetes who participated in this study had been screened. While no specific studies examine why diabetic foot screening is often overlooked, several factors may contribute to this issue. These can include the high workload of primary healthcare (PHC) staff, along with a lack of resources and insufficient knowledge about how to assess diabetic foot conditions [43,45,46]

Interestingly, follow-up questioning revealed that 37% of these patients were only asked about their foot problems. In this study, 17% of persons with diabetes were classified as high-risk; these patients had never received risk stratification before. High-risk patients can be identified through a history of previous ulceration or amputation and foot screening, encompassing impaired sensation, absent pedal pulse, calluses, foot deformities, and inappropriate footwear. This should be a concern, as a history of DFU is the strongest predictor of future foot ulceration, with a one-year recurrence rate of 40% and a recurrence rate of approximately 65% at 5 years [12,15]. The 5-year mortality rate for individuals with diabetic foot ulcers is approximately 30% and exceeds 70% for those with major amputations [13].

The findings of this study regarding suboptimal diabetic foot screening practices at primary healthcare (PHC) facilities align with previous research conducted in South Africa [21,47]. However, a critical distinction must be made; most earlier studies relied heavily on retrospective patient file audits, which primarily highlighted omissions in documentation and screening practices [4,22,48]. While these studies have been instrumental in identifying systemic gaps, they have offered limited insight into the actual clinical presentation and risk profiles of patients with diabetes at risk of developing DFUs. Only a small number of studies have involved direct foot assessments, and notably, these were conducted over a decade ago, with minimal follow-up research since.

Where diabetic foot risk factors have been documented, such data have predominantly originated from tertiary-level facilities [31,32,33]. This has resulted in a significant evidence gap at the PHC level, where most patients with diabetes first seek care. The absence of empirical data on presenting risk factors and patient risk stratification at PHC clinics means that the burden of diabetic foot complications remains largely unknown. The lack of empirical data on presenting diabetic foot risk factors and patient risk stratification at the PHC level has long obscured the true burden of diabetic foot complications in South Africa. This knowledge gap limits clinical insight and weakens the foundation for effective policy direction. Without accurate risk profiling, healthcare providers cannot make informed decisions regarding referral urgency, follow-up intensity, and the allocation of preventive resources, factors critical to averting severe outcomes such as ulceration and amputation.

This study breaks new ground by employing direct clinical assessments within PHC facilities, offering a methodological shift from the retrospective file audits that have dominated previous research. By capturing real-time clinical data and stratifying patients into risk categories, the study introduces a more precise and actionable framework for understanding diabetic foot risk at the point of care. This approach enhances the potential for early intervention, targeted education, and timely management strategies—elements essential to improving patient outcomes and reducing the long-term burden of diabetic foot disease.

Findings revealed that more than half of the patients assessed had one or more risk factors for diabetic foot complications, including peripheral neuropathy and non-palpable pulses, foot deformities such as dropped metatarsal heads, pes cavus, hallux valgus and hyperkeratosis. These results confirm the inadequacy of current screening practices and highlight the urgent need for routine, structured foot screenings at the PHC level.

The successful implementation of diabetic foot screening programmes at the primary healthcare (PHC) level is crucial for ensuring timely access to specialist services. This requires strong communication between primary and secondary care providers. However, the limited number of podiatrists at tertiary care facilities and the lack of structured high-risk foot care teams create significant barriers to delivering integrated and responsive care for patients at high risk. These systemic challenges highlight the urgent need for investment in multidisciplinary foot care services and improved referral systems to facilitate early intervention and prevent complications. Despite these limitations, healthcare providers at the PHC level refer patients at high risk to tertiary levels of care where they are managed accordingly.

To improve early intervention, the study’s findings on risk stratification must be paired with efforts to strengthen PHC systems. It is essential to support PHC staff in navigating existing referral pathways that direct patients from primary care to more specialised secondary and tertiary care. However, challenges such as the lack of dedicated high-risk foot clinics, a limited number of podiatrists, and restricted access to vascular and orthopaedic specialists make it difficult for patients to receive timely care. Currently, the number of podiatrists employed by the state sector in Gauteng province is 61 [49]. To address these issues, it is crucial to implement clearer referral protocols, provide continuous staff training, integrate specialists like podiatrists into care teams, and allocate targeted funding. These steps are vital for effectively managing diabetic foot risk at all levels of care.

## 5. Conclusions

This study represents a deliberate change in approach from reviewing past medical records to conducting direct clinical assessments of diabetic foot risk factors at the primary healthcare level. Focusing on real-time evaluations provides a more accurate and actionable understanding of patients’ clinical presentations, enhancing the accuracy of diabetic foot risk identification at the point of care. The lack of data on diabetic foot risk factors and patient stratification in primary healthcare clinics has made it difficult to understand the extent of diabetic foot complications, hindering clinical decision-making and broader health system planning. This study addressed this gap by collecting real-time data and categorising patients according to risk.

The study’s findings highlight the need, feasibility, and value of incorporating diabetic foot screening and risk stratification into routine primary healthcare workflows. A proactive, patient-centred approach could enable earlier interventions, improve patient education, and strengthen prevention strategies, ultimately aiming to reduce the incidence of diabetic foot ulcers, amputations, and hospital admissions.

The absence of a national diabetes and diabetic foot registry and the limited availability of podiatrists—only 61 in Gauteng’s public sector—severely constrain the ability to monitor diabetic foot outcomes and care quality. These systemic gaps have contributed to slow progress in improving diabetic foot care, despite growing awareness. To accelerate change, policy-driven research is urgently needed to generate actionable evidence to inform strategic planning and resource allocation. Comparative longitudinal studies on podiatric interventions and limb salvage are underway, offering a critical opportunity to build the evidence base needed to influence policy and guide the development of more effective, integrated care models. To improve care delivery, it is essential to establish standardised screening protocols, implement consistent risk stratification, and provide targeted training for healthcare workers. Embedding these practices within primary healthcare settings could allow for the timely identification of high-risk individuals and facilitate early intervention.

The study demonstrates that strengthening diabetic foot screening at the primary healthcare level is practical and urgent. It offers a scalable opportunity to reduce the burden of diabetic foot disease, improve patient outcomes, and promote equity in chronic disease management. By incorporating these practices into routine care, South Africa can move towards a more proactive, sustainable, and patient-centred model of diabetes foot care.

### Study Limitations

This study was observational in nature, which limits the ability to establish causal relationships. Although the researcher collected data on key demographic and clinical variables (e.g., age, sex, duration of diabetes, and comorbidities) to account for potential confounders, only bivariate analyses were performed to explore these associations. A multivariate logistic regression model was not implemented because of sample size constraints, which may have reduced the statistical power and increased the risk of residual confounding. These limitations should be considered when interpreting the findings, and future studies with larger samples are recommended to apply multivariate modelling for a more robust adjustment of confounding factors.

## Figures and Tables

**Table 1 ijerph-22-01794-t001:** Demographic characteristics of the participants (N = 597).

Variable	Type	Frequency	%	*p*-Value
Population group	Black	411	69%	NC
	Asian	30	5%	NC
	Mixed-race	72	12%	NC
	White	84	14%	NC
Sex	Male	263	44%	0.003
	Females	334	56%	0.281
Duration of diabetes	≤1 year	36	6%	0.251
	1–5 years	78	13%	0.575
	6–10 years	269	45%	0.065
	11–15 years	96	16%	0.002
	16–20 years	88	15%	0.055
	20+ years	30	5%	0.061

Table legend: NC—no comparison made.

**Table 2 ijerph-22-01794-t002:** Multimorbidity, Comorbidities and Random blood glucose levels.

Variable	Type	Frequency	%	*p*-Value
Multimorbidity ^1^	Diabetes only	209	35%	0.021
	Diabetes + 1	191	32%	
	Diabetes + 2	137	23%	
	Diabetes + 3	60	10%	
Presenting Comorbidity	Hypertension	373	63%	NC
	Hypercholesterolemia	194	33%	NC
	Chronic Kidney Disease	32	5%	NC
	Cardiac Disease	28	5%	NC
	Retroviral Disease	45	8%	NC
	Obese	179	30%	NC
Random Blood Glucose	0–5 mmol/L	54	9%	0.095
	6–10 mmol/L	346	58%	0.067
	11–15 mmol/L	125	21%	0.003
	16–20 mmol/L	60	10%	NC
	21–25 mmol/L	12	2%	NC

Table legend: NC—no comparisons made. ^1^ Multimorbidity in this study appears to be associated with female sex and increasing age, though this was not investigated further.

**Table 3 ijerph-22-01794-t003:** Recorded diabetic foot risk factors.

Risk Factor	Condition	Frequency	%	*p*-Value
Foot ulcer (n = 597)	Current	116	19%	0.017
	In remission	106	18%	NC
	Never	375	data	NC
Number of ulcers per foot (n = 116)	1	88	76%	0.575
	≥2	28	24%	0.286
Ulcer site (n = 116)	Toes	13	11%	0.286
	Plantar metatarsal area	85	75%	0.017
	Medial longitudinal arch	3	3%	0.286
	Heel	15	13%	0.286
History of amputation (n = 597)	Yes	89	15%	0.713
	No	508	85%	0.672
Type of amputation (n = 103)	Transmetatarsal	41	40%	0.910
	Ankle	31	30%	0.575
	Transtibial	29	28%	0.575
	Transfemoral	2	2%	0.575
Derm Findings (n = 597)	Hyperkeratosis	239	40%	0.812
	Fissures	191	32%	0.281
	Fungal Infection	149	25%	0.286
	Interdigital maceration	125	21%	0.612
Neuropathy (n = 597)	Numbness (LOPS)	198	33%	0.017
	Burning	95	16%	0.286
	Paraesthesia	167	28%	0.286
Peripheral Arterial Disease (n = 597)	Pulses palpable	345	58%	0.681
	Pulses not palpable	131	22%	0.037
	Pulses palpable but faint	125	21%	0.072
Foot deformities (n = 597)	Pes cavus	149	25%	0.041
	Pes planus	30	5%	0.561
	Prominent metatarsal heads	185	31%	0.002
	Hallux valgus	125	21%	0.612

Table legend: NC—no comparisons made.

**Table 4 ijerph-22-01794-t004:** Diabetic foot screening history.

Diabetic Foot Screening in the Past 12 Months		Frequency	%	*p*-Value
Diabetic Foot Screening (n = 597)	Yes	60	10%	NC
	No	537	90%	NC
Nature of screening (n = 115)	HCP looked at my feet	38	63%	NC
	HCP only asked about my feet	22	37%	NC
Type of screening (n = 38)	Skin	14	37%	NC
	Nails	4	10%	NC
	Deformity	5	13%	0.984
	Temparature	6	16%	NC
	Sensation	6	16%	NC
	Footwear	3	8%	NC

Table legend: NC—no comparisons made.

**Table 5 ijerph-22-01794-t005:** Risk stratification of the study participants.

Risk Stratification (n = 597)	Frequency	%
0—Very Low	178	30%
1—Low	202	34%
2—Moderate	116	19%
3—High	101	17%

## Data Availability

Data will be made available on reasonable request following the receipt of authorisation from the author’s local ethics committee.

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
