# Peer review of "Int. J. Environ. Res. Public Health2025, 22(12), 1794;https://doi.org/10.3390/ijerph22121794"

_ijerph, 2025, doi:10.3390/ijerph22121794_

Round 1

Reviewer 1 Report (New Reviewer)

Comments and Suggestions for Authors

The article is relevant and timely. It gives real-time data on both the presence and nature of diabetic foot risk factors and the stratification of patients into risk categories through direct clinical assessments. The paper is well structured, and the results and conclusions match the stated goals. It would be useful to clarify in more detail what specific possibilities are available, once the risk is identified, in terms of multidisciplinary work and other support within these institutions. It would also be valuable to provide more detail on the results of diabetic foot stratification 3.4. After these corrections I recommend the article for publication.

Author Response

Reviewer 1

The article is relevant and timely. It gives real-time data on both the presence and nature of diabetic foot risk factors and the stratification of patients into risk categories through direct clinical assessments. The paper is well structured, and the results and conclusions match the stated goals.

It would be useful to clarify in more detail what specific possibilities are available, once the risk is identified, in terms of multidisciplinary work and other support within these institutions.

I have added the following: The successful implementation of diabetic foot screening programs at the primary healthcare (PHC) level is crucial for ensuring timely access to specialist services. This requires strong communication between primary and secondary care providers. However, the limited number of podiatrists at tertiary care facilities, along with the lack of structured high-risk foot care teams, creates significant barriers to delivering integrated and responsive care for patients at high risk. These systemic challenges highlight the urgent need for investment in multidisciplinary foot care services and improved referral systems to facilitate early intervention and prevent complications

It would also be valuable to provide more detail on the results of diabetic foot stratification 3.4.

I have added the following: Risk stratification of the study cohort using the IWGDF classification system revealed that 30% (178 out of 597) of patients were categorised as having a very low risk, and 17% were identified as high risk. Importantly, patients in the high-risk category presented with active foot pathology, indicating a significantly increased likelihood of complications such as ulceration or infection. These findings illustrate the distribution of risk levels within the cohort and provide a basis for targeted clinical interventions aligned with individual patient risk profiles.

After these corrections, I recommend the article for publication.

Reviewer 2 Report (New Reviewer)

Comments and Suggestions for Authors

The topic of diabetes treatment is important for the entire global community, and the author's initiative deserves respect.

However, there are some comments/observations:

  1. The article provides absolute numbers for the number of patients, but does not provide the overall percentage of patients with diabetes, so it is impossible to understand how serious the problem is for South Africa (or the Republic of South Africa, which is more familiar to many readers).
  2. The manuscript does not distinguish between type 1 and type 2 diabetes, which would be useful to indicate (this is a fundamental issue, including in terms of treatment).
  3. There is no mention of patient adherence to treatment, although if, as the author indicates, they regularly visit outpatient clinics to monitor their glucose and glycated hemoglobin levels, it is reasonable to assume that adherence is high. However, it would be beneficial to include this aspect of treatment adherence in the manuscript.
  4. It would be interesting to provide correlations of the foot condition with other objective indicators determined during an outpatient visit and presented in the tables. If it turns out that the information "from the foot" correlates more with the severity and outcomes of diabetes, this would serve as scientific evidence for the need to change the protocol. After all, all the signs mentioned in the tables already correlate with the severity and outcomes of diabetes.
  5. It would be useful for both the reader and healthcare professionals to understand the reason for the lack of sufficient attention (from the author's perspective) to assessing the foot condition. Could it be due to a simple lack of time (20 minutes spent by the author conducting the research)?
  6. The author has a whole series of articles on the issue under discussion. The first article dates 2016(?). The same idea is repeated throughout the series, which is that an outpatient examination of the diabetic foot can improve treatment outcomes, including reducing the number of amputations. It is possible that the author is a podologist. This idea is not new, and the author constantly promotes it, but it is unclear what prevents the author from conducting a comparative analysis of the course of the disease based on these two criteria using the example of South Africa and providing data that is significant for healthcare organizations, which would lead to a regulatory change in the management of patients with diabetes. It would be useful to present new comparative data on the results before and after the author's proposed approach to the treatment of diabetic foot. What will this change? How many legs will be saved? How will it reduce the disability?
  7. References are incorrectly formatted. There is no DOI. Self-citation – 3 works out of 38 (No. 20, 25, and 35, with no byline in the 35th work).

Author Response

Response to Reviewer 1 Comments

1. Summary

2. Questions for General Evaluation

Reviewer’s Evaluation

Response and Revisions

Does the introduction provide sufficient background and include all relevant references?

Yes/Can be improved/Must be improved/Not applicable

[Please give your response if necessary. Or you can also give your corresponding response in the point-by-point response letter. The same as below]

Are all the cited references relevant to the research?

Yes/Can be improved/Must be improved/Not applicable

Is the research design appropriate?

Yes/Can be improved/Must be improved/Not applicable

Are the methods adequately described?

Yes/Can be improved/Must be improved/Not applicable

Are the results clearly presented?

Yes/Can be improved/Must be improved/Not applicable

Are the conclusions supported by the results?

Yes/Can be improved/Must be improved/Not applicable

3. Point-by-point response to Comments and Suggestions for Authors

  1. Comments 1: [The article provides absolute numbers for the number of patients, but does not provide the overall percentage of patients with diabetes, so it is impossible to understand how serious the problem is for South Africa (or the Republic of South Africa, which is more familiar to many readers).]

Response 1: Thank you for pointing this out. I have indicated the percentage where relevant, along with absolute numbers.

  1. Comments 2: [The manuscript does not distinguish between type 1 and type 2 diabetes, which would be useful to indicate (this is a fundamental issue, including in terms of treatment).]

Response 2: Agree. I have indicated that the study was on patients with type 2 diabetes.

  1. Comments 3: There is no mention of patient adherence to treatment, although if, as the author indicates, they regularly visit outpatient clinics to monitor their glucose and glycated hemoglobin levels, it is reasonable to assume that adherence is high. However, it would be beneficial to include this aspect of treatment adherence in the manuscript.
  1. Response 3: Noted the following has been added on the introduction Studies have found that most South Africans with T2D are managed at the PHC level, where the standard of care is inadequate; only 10–30% of patients in the public health system achieve glycaemic control or an HbA1c of< 7.0% [1, 2]. Despite strong evidence supporting intensive blood glucose control, many patients with type 2 diabetes remain on suboptimal therapy. Numerous studies have linked this gap to healthcare professionals not intensifying treatment when clinically indicated, contributing to poor glycaemic outcomes [6]
  1. Comments 4: It would be interesting to provide correlations of the foot condition with other objective indicators determined during an outpatient visit and presented in the tables. If it turns out that the information "from the foot" correlates more with the severity and outcomes of diabetes, this would serve as scientific evidence for the need to change the protocol. After all, all the signs mentioned in the tables already correlate with the severity and outcomes of diabetes.
  1. Response 4: The comment is noted, and the following attempt has been included to capture the correlations noted in the study. This study identified key risk factors associated with diabetic foot complications, with significant correlations observed between poor glycaemic control and current ulceration. Patients with random blood glucose levels of 11–15 mmol/L were more likely to present with foot complications (p=0.003), reinforcing the importance of effective glucose management.

  1. Multimorbidity also showed a significant association, particularly among patients with diabetes and two additional chronic conditions (p=0.021), suggesting that cumulative disease burden increases vulnerability. Structural and neurological factors, such as numbness (LOPS), non-palpable pulses, pes cavus, and prominent metatarsal heads, were also significantly linked to foot complications. These findings highlight the need for integrated screening and management strategies and underscore the urgency for policy-driven research to strengthen diabetic foot care pathways and inform resource allocation.
  2.  
  1. Comments 5: It would be useful for both the reader and healthcare professionals to understand the reason for the lack of sufficient attention (from the author's perspective) to assessing the foot condition. Could it be due to a simple lack of time (20 minutes spent by the author conducting the research)?
  1. Response 5: Thank you for this comment. I have added the following with supporting literature. However, it is important to keep in mind that there could be systemic challenges faced by healthcare professionals at PHC facilities in delivering quality diabetic foot care. These include limited resources, insufficient training among healthcare workers, and a lack of standardised screening protocols and referral pathways [3, 4]. Nurses working at PHC facilities see between 40 and 50 patients a day, which limits their consultation times with each patient [5, 6].

  1. Comments 6: The author has a whole series of articles on the issue under discussion. The first article dates to 2016(?). The same idea is repeated throughout the series, which is that an outpatient examination of the diabetic foot can improve treatment outcomes, including reducing the number of amputations. It is possible that the author is a podologist. This idea is not new, and the author constantly promotes it, but it is unclear what prevents the author from conducting a comparative analysis of the course of the disease based on these two criteria using the example of South Africa and providing data that is significant for healthcare organizations, which would lead to a regulatory change in the management of patients with diabetes. It would be useful to present new comparative data on the results before and after the author's proposed approach to the treatment of diabetic foot. What will this change? How many legs will be saved? How will it reduce the disability?

2.       Response 6: Thank you for the above comments. Let me begin with a personal note that is not included in the manuscript. The first work you mentioned was my Master’s thesis, which looked at the need for the inclusion of podiatrists in the PHC team. One of the key challenges in the Republic of South Africa is the absence of structured podiatry services, a dedicated foot health policy, and integrated health service frameworks. This is further compounded by a generally limited commitment to research and publication within the podiatry profession. I am currently working on a position statement to address these gaps. In the meantime, however, the status quo remains unchanged.

 I have done my best to respond comprehensively. I have included the following in the discussion: To improve early intervention, the study's findings on risk stratification must be paired with efforts to strengthen PHC systems. It is essential to support PHC staff in navigating existing referral pathways that direct patients from primary care to more specialised secondary and tertiary care. However, challenges such as the lack of dedicated high-risk foot clinics, a limited number of podiatrists, and restricted access to vascular and orthopaedic specialists make it difficult for patients to receive timely care. Currently, the number of podiatrists employed by the state sector in Gauteng province is 61 [7]. To address these issues, it is crucial to implement clearer referral protocols, provide continuous training for staff, integrate specialists like podiatrists into care teams, and allocate targeted funding. These steps are vital for effectively managing diabetic foot risk at all levels of care.

  1. I have also included the following in the conclusion: The absence of a national diabetes and diabetic foot registry and the limited availability of podiatrists—only 61 in Gauteng’s public sector- severely constrain the ability to monitor diabetic foot outcomes and care quality. These systemic gaps have contributed to slow progress in improving diabetic foot care, despite growing awareness. To accelerate change, there is an urgent need for policy-driven research that generates actionable evidence to inform strategic planning and resource allocation. Encouragingly, comparative longitudinal studies on podiatric interventions and limb salvage are currently underway, offering a critical opportunity to build the evidence base needed to influence policy and guide the development of more effective, integrated care models. To improve care delivery, it is essential to establish standardised screening protocols, implement consistent risk stratification, and provide targeted training for healthcare workers. Embedding these practices within primary healthcare settings could allow for the timely identification of high-risk individuals and facilitate early intervention.

Comments 7: References are incorrectly formatted. There is no DOI. Self-citation – 3 works out of 38 (No. 20, 25, and 35, with no byline in the 35th work)

2. Response 7: This is noted and has been amended accordingly.

4. Response to Comments on the Quality of English Language

Point 1:

Response 1:    (in red)

5. Additional clarifications

[Here, mention any other clarifications you would like to provide to the journal editor/reviewer.]

Reviewer 3 Report (New Reviewer)

Comments and Suggestions for Authors

Thank you for the opportunity to review this paper. This is a really important area to study, with results indicating the need for increased foot assessment and risk stratification in primary care. The article is very well written.

Some comments for consideration by the authors.

  1. I suggest consideration of the Diabetes Language Position statement - Language-Matters-Diabetes-Australia-Position-Statement.pdf - particularly to avoid terms like "diabetic" / "treatment" (e.g "diabetic treatment guidelines" - replace with something like "guidelines for the management of diabetes") "diabetic patient" (replace with "person with diabetes" for example). (Our language matters: Improving communication with and about people with diabetes. A position statement by Diabetes Australia - Diabetes Research and Clinical Practice)
  2. In the abstract - query some inaccurate data. n=597, 33% with neuropathy (1498). 33% of 597 would be 197 - so just querying this.
  3. I wonder if there are references to support some statements in the intro (e.g. Patients with diabetes are more at risk of developing foot problems, with those affected experiencing higher rates of foot ulceration, lower-limb amputation and premature death / These ulcers are primarily preceded by risk factors, including peripheral neuropathy, ischaemia, and foot deformities, and are the leading cause of lower-extremity amputations.). There are a number of statements throughout the paper that aren't supported by any evidence or the current study's findings.
  4. Use full words on first use of acronym. e.g. line 45 DFU - first explanation of the acronym is in line 57.
  5. Line 190 typo - the was a significant association - 'there' was
  6. In the discussion line 227 the following is mentioned: "However, the results reveal that this potential is not being fully realised due to the absence of structured, institutionalised screening protocols." - during the data collection period was each community health service questioned about their screening protocols? This information isn't presented - so perhaps they have protocols, but they aren't being followed, or staff aren't educated on them / staff are not confident or skilled to perform a foot check or risk assessment. I just query this statement (as it's not a direct finding from the results), and wonder if there is more to it.
  7. Line 242 - "Most diabetic foot complications can be prevented through early screening". - is there evidence for this statement?
  8. Line 247 - evidence from THIS study indicates that diabetes foot screening is often neglected or are you referring to other studies? If other studies, need to cite these. 
  9. Line 249 - you've identified some factors that impact screening rates - did you have a citation for these also? Need to ensure your statements are supported by evidence. Having a bit more of a focus on this area may add value - just 10% of participants had a foot screen in the last 12 months - what does the literature say about why this might be happening? Further, with the risk stratification completed - what procedures/policies are in place for PHCs to respond to these risk levels and then are the clinics/staff supported and resourced for this - e.g what's the current funding/educational levels, are there high risk foot clinics/specialist podiatrists/vascular/orthos available and accessible. - just some thoughts for consideration by the authors. 

Author Response

Response to Reviewer 2 Comments

Thank you very much for taking the time to review this manuscript. Please find detailed responses below and the corresponding revisions/corrections highlighted/in track changes in the re-submitted files.

Review 2

2. Questions for General Evaluation

Reviewer’s Evaluation

Response and Revisions

Does the introduction provide sufficient background and include all relevant references?

Yes/Can be improved/Must be improved/Not applicable

[Please give your response if necessary. Or you can also give your corresponding response in the point-by-point response letter. The same as below]

Are all the cited references relevant to the research?

Yes/Can be improved/Must be improved/Not applicable

Is the research design appropriate?

Yes/Can be improved/Must be improved/Not applicable

Are the methods adequately described?

Yes/Can be improved/Must be improved/Not applicable

Are the results clearly presented?

Yes/Can be improved/Must be improved/Not applicable

Are the conclusions supported by the results?

Yes/Can be improved/Must be improved/Not applicable

3. Point-by-point response to Comments and Suggestions for Authors

  1. Comments 1: [I suggest consideration of the Diabetes Language Position statement - Language-Matters-Diabetes-Australia-Position-Statement.pdf - particularly to avoid terms like "diabetic" / "treatment" (e.g "diabetic treatment guidelines" - replace with something like "guidelines for the management of diabetes") "person with diabetes" (replace with "person with diabetes" for example). (Our language matters: Improving communication with and about people with diabetes. A position statement by Diabetes Australia - Diabetes Research and Clinical Practice).]

Response 1: [Type your response here and mark your revisions in red.] Thank you for pointing this out. I/have adjusted these terms to reflect the proper language as indicated.

  1. Comments 2: [In the abstract - query some inaccurate data. n=597, 33% with neuropathy (1498). 33% of 597 would be 197 - so just querying this.]
  2.  

Response 2: Thank you for picking this up and I agree, I have corrected the number to 197

  1. Comments 3: I wonder if there are references to support some statements in the intro (e.g. Patients with diabetes are more at risk of developing foot problems, with those affected experiencing higher rates of foot ulceration, lower-limb amputation and premature death / These ulcers are primarily preceded by risk factors, including peripheral neuropathy, ischaemia, and foot deformities, and are the leading cause of lower-extremity amputations.). There are a number of statements throughout the paper that aren't supported by any evidence or the current study's findings.

  1. Response: Thank you for pointing out this omission.  I have gone through the document to ensure that all identified statements are referenced accordingly.
  2.  
  1. Comments 4: Use full words on first use of acronym. e.g. line 45 DFU - first explanation of the acronym is in line 57.

  1. Response: The first explanation of the acronym DFU is now included in line 45.

  1. Comments 5: Line 190 typo - the was a significant association - 'there' was

  1. Response: Noted with thanks, and this has been emended accordingly.
  1. Comments 6: In the discussion line 227 the following is mentioned: "However, the results reveal that this potential is not being fully realised due to the absence of structured, institutionalised screening protocols." - during the data collection period was each community health service questioned about their screening protocols? This information isn't presented - so perhaps they have protocols, but they aren't being followed, or staff aren't educated on them / staff are not confident or skilled to perform a foot check or risk assessment. I just query this statement (as it's not a direct finding from the results), and wonder if there is more to it.
  1. Response: Thank you for this. In this study, healthcare professionals were not questioned. In line with this, I have amended this sentence to read as follows: “The author suggests that this potential is not being fully realised, possibly due to the lack of structured, institutionalised screening protocols.” I hope this will show that this was not a direct finding, but rather an observation by the researcher.
  1. Comments 7: Line 242 - "Most diabetic foot complications can be prevented through early screening". - is there evidence for this statement?
  1. Response: Reference has been added
  1. Comments 8: Line 247 - evidence from THIS study indicates that diabetes foot screening is often neglected or are you referring to other studies? If other studies, need to cite these. 
  1. Response: Noted accordingly, the word from this study has been added to remove ambiguity.
  1. Comments 9: Line 249 - you've identified some factors that impact screening rates - did you have a citation for these also? Need to ensure your statements are supported by evidence. Having a bit more of a focus on this area may add value - just 10% of participants had a foot screen in the last 12 months - what does the literature say about why this might be happening? Further, with the risk stratification completed, what procedures/policies are in place for PHCs to respond to these risk levels and then are the clinics/staff supported and resourced for this - e.g what's the current funding/educational levels, are there high risk foot clinics/specialist podiatrists/vascular/orthos available and accessible. - just some thoughts for consideration by the authors. 
  1. Response 9: The points raised here are well taken. To improve this, I have added the following sentences. To address the comment on what the literature says about low screening practices, I have added. “While there are no specific studies examining why diabetic foot screening is often overlooked, several factors may contribute to this issue. These can include the high workload of primary healthcare (PHC) staff, along with a lack of resources and insufficient knowledge about how to assess diabetic foot.

To address the point on how findings could be useful, I have added: To improve early intervention, the study's findings on risk stratification must be paired with efforts to strengthen PHC systems. It is essential to support PHC staff in navigating existing referral pathways that direct patients from primary care to more specialised secondary and tertiary care. However, challenges such as the lack of dedicated high-risk foot clinics, a limited number of podiatrists, and restricted access to vascular and orthopaedic specialists make it difficult for patients to receive timely care. To address these issues, it is crucial to implement clearer referral protocols, provide continuous training for staff, integrate specialists like podiatrists into care teams, and allocate targeted funding. These steps are vital for effectively managing diabetic foot risk at all levels of care.

4. Response to Comments on the Quality of English Language

Point 1:

Response 1:    (in red)

5. Additional clarifications

[Here, mention any other clarifications you would like to provide to the journal editor/reviewer.]

All reviews related to reviewer 3 are also highlighted in green in the revised manuscript.

Round 2

Reviewer 2 Report (New Reviewer)

Comments and Suggestions for Authors

After reviewing the revised manuscript and the authors' detailed responses, I consider all concerns raised during the peer review process to have been fully addressed. I therefore recommend the manuscript for publication.

Author Response

After reviewing the revised manuscript and the authors' detailed responses, I consider all concerns raised during the peer review process to have been fully addressed. I therefore recommend the manuscript for publication.

Noted with appreciation.

This manuscript is a resubmission of an earlier submission. The following is a list of the peer review reports and author responses from that submission.

Round 1

Reviewer 1 Report

Comments and Suggestions for Authors

Section 2.5 (Data Analysis): It needs to be clearly specified which statistical tests were performed for the analyses presented. Currently, the statistical test used to obtain the p-values is not indicated. Pages 135-140.

Section 3 (Results - Table 2): In Table 2, an asterisk is observed next to the "Multimorbidity *" variable. A legend or footnote explaining the meaning of this symbol needs to be included.

Section 3 (Results - Tables 1, 2, 3 and 4): Some p values for some variables in Tables 1, 2, 3 and 4, are missing. It should be explained somewhere in the results section, what is their explanation. No comparisons were made? Justify.

Section 3 (Results - Line 159): The figure of “185” on line 159 (“Thirty-one percent (31%, 185/597) presented with neuropathy”) does not appear to correspond to the values presented in Table 3 for neuropathy. Please clarify this data.

Line 159 reads: “Thirty-one percent (31%, 185/597) presented with neuropathy”. What is the origin of the 185? It does not correspond with the values in Table 3.

Section References: 15 and 23 are duplicated in the reference list. A review and verification of all citations throughout the text is requested.

Author Response

Reviewer 1

Section 2.5 (Data Analysis): It needs to be clearly specified which statistical tests were performed for the analyses presented. Currently, the statistical test used to obtain the p-values is not indicated. Pages 135-140.

Continuous data are summarised as mean ± standard deviation or median (interquartile ranges), while categorical data are expressed as frequencies and percentages. The Chi-square test was used to test for associations between categorical variables.

Section 3 (Results - Table 2): In Table 2, an asterisk is observed next to the "Multimorbidity *" variable. A legend or footnote explaining the meaning of this symbol needs to be included.

The following footnote, initially missed, has been included: Multimorbidity in this study appears to be associated with female sex and increasing age, though this was not investigated further.

Section 3 (Results - Tables 1, 2, 3 and 4): Some p values for some variables in Tables 1, 2, 3 and 4, are missing. It should be explained somewhere in the results section, what is their explanation. No comparisons were made? Justify.

This has been updated as necessary. Where no comparisons were made, I have indicated this; accordingly, please refer to the revised tables.

Section 3 (Results - Line 159): The figure of “185” on line 159 (“Thirty-one percent (31%, 185/597) presented with neuropathy”) does not appear to correspond to the values presented in Table 3 for neuropathy. Please clarify this data.

Noted and corrected, this was an error in mixing up the 185/597 (31%) for prominent metatarsals instead of the 198/597 (33%) for identified numbness LOPS.

Line 159 reads: “Thirty-one percent (31%, 185/597) presented with neuropathy”. What is the origin of the 185? It does not correspond with the values in Table 3.

Noted and corrected, this was an error in mixing the 185/597 (31%) for prominent metatarsals instead of the 198/597 (33%) for identified LOPS. The origin of the 185 is the mix-up with the value of the prominent metatarsals.

Section References: 15 and 23 are duplicated in the reference list. A review and verification of all citations throughout the text is requested.

Reviewer 2 Report

Comments and Suggestions for Authors

Thank you for the opportunity to review this manuscript, which addresses a highly relevant clinical and public health issue, particularly in the South African context. The growing burden of diabetic foot complications and the limited implementation of screening practices in primary care justify the need for studies like this one. The work is timely and provides valuable evidence on deficiencies in the early detection of these complications at the primary healthcare level.

However, the manuscript presents a series of methodological limitations, inconsistencies in the description of the study, and important omissions that must be addressed before it can be considered for publication. First, there is a discrepancy in the number of participating centers reported: the abstract mentions five community centers, while the Methods section states that the study was conducted in ten. This kind of inconsistency raises concerns about the study’s internal consistency and should be corrected with precision.

The use of convenience sampling also represents a significant limitation. While this is understandable given the operational nature of the study in community clinics, it should be more clearly justified and, more importantly, discussed in terms of its implications for the external validity of the results. As currently written, the manuscript does not adequately address this issue. Additionally, it is unclear whether any control for confounding factors was performed. Although significant associations are reported between various clinical variables (e.g., neuropathy and ulceration), no multivariate analysis is presented. In observational studies of this kind, applying logistic regression models or similar methods is essential to identify independent factors associated with outcomes.

Another aspect that requires improvement is the description of the data collection instrument. It is mentioned that the questionnaire was adapted from a master’s thesis, but no details are provided about its psychometric properties, nor is the instrument included as supplementary material. This is important, as much of the clinical interpretation of the data relies on this tool. I suggest that the author provide the full questionnaire and describe in greater detail the validation process, even if this was only preliminary in the pilot study mentioned.

Regarding the presentation of results, while the structure is clear, there are several typographical and grammatical errors that should be corrected (e.g., “The was a significant association…”). Some discrepancies in reported figures are also noticeable. For instance, the description of participants’ sex states that 58% were female, but the corresponding table reports 56%. These minor errors, although easily correctable, impact confidence in the overall consistency of the analysis.

The discussion is generally appropriate in intent but tends to be repetitive in several sections. Ideas already presented in the introduction or results are reiterated, which weakens the argument. The discussion would benefit from a more concise approach, focusing on contrasting findings with prior literature and emphasizing the practical implications for healthcare policy and clinical practice in primary care. A specific limitations section is also missing. Given the cross-sectional nature of the study, the use of non-random sampling, and the absence of longitudinal follow-up, these limitations must be clearly acknowledged and discussed.

The conclusion is well-aligned with the findings and appropriately outlines recommendations. However, it would be beneficial to strengthen the message regarding the need for healthcare worker training and the institutionalization of diabetic foot screening protocols in primary care centers.

Author Response

Reviewer 2

  • Thank you for the opportunity to review this manuscript, which addresses a highly relevant clinical and public health issue, particularly in the South African context. The growing burden of diabetic foot complications and the limited implementation of screening practices in primary care justify the need for studies like this one. The work is timely and provides valuable evidence on deficiencies in the early detection of these complications at the primary healthcare level.
  • However, the manuscript presents a series of methodological limitations, inconsistencies in the description of the study, and important omissions that must be addressed before it can be considered for publication. First, there is a discrepancy in the number of participating centers reported: the abstract mentions five community centers, while the Methods section states that the study was conducted in ten. This kind of inconsistency raises concerns about the study’s internal consistency and should be corrected with precision.
  • Noted and appreciated, this error has been corrected to indicate that the study was conducted in five centres.
  • The use of convenience sampling also represents a significant limitation. While this is understandable given the operational nature of the study in community clinics, it should be more clearly justified and, more importantly, discussed in terms of its implications for the external validity of the results. As currently written, the manuscript does not adequately address this issue. Additionally, it is unclear whether any control for confounding factors was performed. Although significant associations are reported between various clinical variables (e.g., neuropathy and ulceration), no multivariate analysis is presented. In observational studies of this kind, applying logistic regression models or similar methods is essential to identify independent factors associated with outcomes.
  • An attempt to justify and indicate the implications of the chosen sampling method for the study is now included. In hindsight, the omission of the multivariate analysis is acknowledged. The following statement has been included in the manuscript: The researcher used a convenience sampling method to recruit participants from community clinics. This approach was chosen because operational constraints in these settings made random sampling impractical without disrupting routine clinical workflows and patient care. Convenience sampling allowed the researcher to include participants who were easily accessible during the study period, enabling timely data collection despite the project's resource and time limitations. While this method facilitated practical implementation, the researcher recognises that it may introduce selection bias and limit the representativeness of the sample. The researcher alludes to these implications further in the Limitations section.
  • Another aspect that requires improvement is the description of the data collection instrument. It is mentioned that the questionnaire was adapted from a master’s thesis, but no details are provided about its psychometric properties, nor is the instrument included as supplementary material. This is important, as much of the clinical interpretation of the data relies on this tool. I suggest that the author provide the full questionnaire and describe in greater detail the validation process, even if this was only preliminary in the pilot study mentioned.
  • I have included the following statement in line with how the original questionnaire was validated. The questionnaire used in this study was adapted from an instrument that had been validated during a previous Master’s research project. The validation process included expert review to establish content validity, participant feedback for face validity, and reliability testing using Cronbach’s alpha (α = 0.82), indicating good internal consistency. For the current study, the adapted questionnaire was piloted with ten participants to assess clarity, relevance, and feasibility in the intended context. Feedback from the pilot was used to make minor refinements, ensuring the tool was appropriate for the main study. The questionnaire used to collect data for this study is now included.
  • Regarding the presentation of results, while the structure is clear, there are several typographical and grammatical errors that should be corrected (e.g., “The was a significant association…”). Some discrepancies in reported figures are also noticeable. For instance, the description of participants’ sex states that 58% were female, but the corresponding table reports 56%. These minor errors, although easily correctable, impact confidence in the overall consistency of the analysis.
  • The comment is noted with apology. These grammatical errors have been corrected in the manuscript.
  • The discussion is generally appropriate in intent but tends to be repetitive in several sections. Ideas already presented in the introduction or results are reiterated, which weakens the argument. The discussion would benefit from a more concise approach, focusing on contrasting findings with prior literature and emphasizing the practical implications for healthcare policy and clinical practice in primary care. A specific limitations section is also missing. Given the cross-sectional nature of the study, the use of non-random sampling, and the absence of longitudinal follow-up, these limitations must be clearly acknowledged and discussed.
  • Noted with appreciation. An attempt has been made to present a more concise discussion with contrasting literature to highlight the practical implications. A section on limitations is now included.
  • The conclusion is well-aligned with the findings and appropriately outlines recommendations. However, it would be beneficial to strengthen the message regarding the need for healthcare worker training and the institutionalization of diabetic foot screening protocols in primary care centers.
  • I have reworked the conclusion to read as follows, in an attempt to incorporate the reviewers' comments. Diabetic foot complications, particularly diabetic foot ulcers (DFUs), represent a significant public health concern due to their prevalence, cost, and impact on patient quality of life. Despite being largely preventable through early identification and routine screening, current practices in many South African primary healthcare (PHC) settings remain inadequate. As the first point of contact for most patients, PHC clinics are ideally positioned to play a central role in the prevention, early detection, and management of diabetic foot complications.

Institutionalising standardised screening protocols and investing in comprehensive training for healthcare workers are critical to improving care delivery. These measures will enable timely risk stratification, appropriate foot care, patient education, and effective referral pathways. Strengthening diabetic foot care within PHC not only reduces the incidence of ulcers and amputations but also contributes to more equitable and sustainable diabetes management across the health system.

Reviewer 3 Report

Comments and Suggestions for Authors The paper address the description of the diabetic foot situation in the south of Africa. This is a topic of big clinical-scientific and social interest. However, is important to mention that the study not provide relevant data that is not already published, even if it analyzed a considerable sample and had the approval of the corresponding ethical committee. Still, there are aspects of the work that should be valued.   The introduction is not well structured and is too much extended. The authors repeat ideas and it seems there is not a proper guide that help the reader to go into the topic in a organized way.   Regarding the methodology, it should be pointed that the authors was not detailed enough when describing it, which makes that the study cannot be replicated with the information provided. Also, we consider that the sub-sections do not reflect the corresponding information, but are mixed. They should be more careful with this aspects.   The results are just simple descriptive data. Probably, with the database, more correlations between the studied variables could be concluded.   The conclusions should be clear, direct and concise answers to the objectives. In this case, they include reflections from the authors.   Regarding the discussion, it should be reformulated taking in consideration all the aspects mentioned before.

Author Response

Reviewer 3

The paper address the description of the diabetic foot situation in the south of Africa. This is a topic of big clinical-scientific and social interest. However, is important to mention that the study not provide relevant data that is not already published, even if it analyzed a considerable sample and had the approval of the corresponding ethical committee. Still, there are aspects of the work that should be valued.  

  • The introduction is not well structured and is too much extended. The authors repeat ideas and it seems there is not a proper guide that help the reader to go into the topic in a organized way.  

To address this, I have restructured the entire introduction section to enhance its flow and smoothly guide the reader into the topic..

  • Regarding the methodology, it should be pointed that the authors was not detailed enough when describing it, which makes that the study cannot be replicated with the information provided. Also, we consider that the sub-sections do not reflect the corresponding information, but are mixed. They should be more careful with this aspects.  
  • The methodology section has been reworked to address the reviewers' comments. The sections have been cleaned up to address the corresponding relevant information.
  • The results are just simple descriptive data. Probably, with the database, more correlations between the studied variables could be concluded.  
  • This is acknowledged, and the table has been cleaned up to reflect adequately where comparisons were made to establish correlations.
  • The conclusions should be clear, direct and concise answers to the objectives. In this case, they include reflections from the authors.  
  • The conclusion has been reworked to include a statement on the author’s reflection: Diabetic foot complications, particularly diabetic foot ulcers (DFUs), represent a significant and preventable burden in diabetes care. Despite their prevalence, screening practices in many South African primary healthcare (PHC) settings remain suboptimal. As the first point of contact for most patients, PHC clinics are ideally positioned to facilitate early detection, risk stratification, and timely referral for diabetic foot complications.
  • Institutionalising standardised screening protocols and investing in targeted healthcare worker training are essential to improving care delivery. This approach leads to improved outcomes for patients with diabetes, helping to prevent ulcers and amputations. Positioning this process within PHC enables the identification of individuals most at risk of foot ulceration, allowing for early intervention and more effective diabetic foot care management.

The author believes that strengthening diabetic foot screening at the PHC level is both a practical and urgent priority. It offers a scalable opportunity to reduce the burden of diabetic foot disease, improve patient outcomes, and promote equity in chronic disease management. By embedding these practices into routine PHC workflows, we can move toward a more proactive, sustainable, and patient-centred model of diabetes care across South Africa.

  • Regarding the discussion, it should be reformulated taking into consideration all the aspects mentioned before.
  • Discussion has been reworked to address the key comments from the reviewer without losing the main thrust of the discussion.

Round 2

Reviewer 2 Report

Comments and Suggestions for Authors

Congratulations to the authors on the work done in this new version of the manuscript. After reviewing the article again, I can say that all the requested corrections have been properly addressed, and the result is a much more solid and clear piece of work. The quality of the content has improved significantly, and the study is now presented with greater coherence and depth. It is a valuable contribution within its field, and in my opinion, it is ready to be published. Well done.

Author Response

Noted and much appreciated for the input to the final work on this manuscript.

Reviewer 3 Report

Comments and Suggestions for Authors

The authors addressed some of the comments made in the review; however, I think that despite these changes, the article was not clarified properly. It is also important to mention again that the study does not provide relevant data that are not already published.

Author Response

The authors addressed some of the comments made in the review; however, I think that despite these changes, the article was not clarified properly.

  • Please note the response regarding the positioning of the study. I have also included risk stratification in the topic to illustrate the nuance of the research and its potential relevance. I hope that this clarifies the study adequately.

This study addresses a significant knowledge gap in the South African diabetic foot care literature. To date, clinical data on diabetic foot complications have primarily been obtained from tertiary-level facilities. Conversely, research at the primary healthcare level has largely relied on retrospective patient file reviews, which are useful for identifying documentation gaps and inconsistent screening practices but offer limited insight into the actual diabetic foot clinical presentation and risk profiles of patients, particularly those presenting with early signs of diabetic foot complications. Notably, no prior studies in South Africa have produced empirical data on the clinical presenting risk factors or the stratification of patients by risk category at the primary healthcare level, representing a substantial shortfall in the evidence base. Presenting diabetic foot risk factors and risk stratification are fundamental to effective diabetic foot prevention, directly informing immediate clinical decisions, including referral urgency, follow-up intensity, and resource allocation. In the absence of such data, policymakers and healthcare providers lack the necessary information to tailor interventions to individual patient risk levels, potentially delaying care and increasing the likelihood of adverse outcomes. This study generated real-time data on both the presence and nature of diabetic foot risk factors and the stratification of patients into risk categories through direct clinical assessments. The findings confirm the inadequacy of current screening practices and provide a foundation for more responsive, risk-informed care. The study's contribution is both timely and novel, offering actionable insights that can inform public health policy, strengthen primary healthcare protocols, and reduce the burden of diabetic foot complications across all levels of care.

 It is also important to mention again that the study does not provide relevant data that are not already published.

  • I have adjusted the discussion and conclusion section slightly to provide clarification or novelty of the study findings in contrast to previous studies. The points on actual patient assessments and risk stratification are made to stand out

Discussion adjustments

The findings of this study regarding suboptimal diabetic foot screening practices at primary healthcare (PHC) facilities align with previous research conducted in South Africa 1, 2. However, a critical distinction must be made; most earlier studies relied heavily on retrospective patient file audits, which primarily highlighted omissions in documentation and screening practices 3-5. While these studies have been instrumental in identifying systemic gaps, they have offered limited insight into the actual clinical presentation and risk profiles of patients with diabetes at risk of developing DFUs. Only a small number of studies have involved direct foot assessments, and notably, these were conducted over a decade ago, with minimal follow-up research since.

 Where diabetic foot risk factors have been documented, such data have predominantly originated from tertiary-level facilities 6-8. This has resulted in a significant evidence gap at the PHC level, where most patients with diabetes first seek care. The absence of empirical data on presenting risk factors and patient risk stratification at PHC clinics means that the true burden of diabetic foot complications remains largely unknown. The lack of empirical data on presenting diabetic foot risk factors and patient risk stratification at the PHC level has long obscured the true burden of diabetic foot complications in South Africa. This knowledge gap not only limits clinical insight but also weakens the foundation for effective policy direction. Without accurate risk profiling, healthcare providers are unable to make informed decisions regarding referral urgency, follow-up intensity, and the allocation of preventive resources—factors that are critical to averting severe outcomes such as ulceration and amputation.

This study breaks new ground by employing direct clinical assessments within PHC facilities, offering a methodological shift from the retrospective file audits that have dominated previous research. By capturing real-time clinical data and stratifying patients into risk categories, the study introduces a more precise and actionable framework for understanding diabetic foot risk at the point of care. This approach enhances the potential for early intervention, targeted education, and timely management strategies—elements essential to improving patient outcomes and reducing the long-term burden of diabetic foot disease.

Findings revealed that more than half of the patients assessed had one or more risk factors for diabetic foot complications, including peripheral neuropathy and non-palpable pulses, foot deformities such as dropped metatarsal heads, pes cavus, hallux valgus and hyperkeratosis. These results confirm the inadequacy of current screening practices and highlight the urgent need for routine, structured foot screenings at the PHC level.

Conclusion

This study represents a deliberate change in approach from reviewing past medical records to conducting direct clinical assessments of diabetic foot risk factors at the primary healthcare level. By focusing on real-time evaluations, it provides a more accurate and actionable understanding of patients' clinical presentations, which in turn enhances the accuracy of diabetic foot risk identification at the point of care. The lack of data on diabetic foot risk factors and patient stratification in primary healthcare clinics has made it difficult to understand the true extent of diabetic foot complications, hindering both clinical decision-making and broader health system planning. This study addressed this gap by collecting real-time data and categorising patients according to risk.

The study's findings highlight the need, feasibility, and value of incorporating diabetic foot screening and risk stratification into routine primary healthcare workflows. A proactive, patient-centred approach could enable earlier interventions, improve patient education, and strengthen prevention strategies, ultimately aiming to reduce the incidence of diabetic foot ulcers, amputations, and hospital admissions.

To improve care delivery, it is essential to establish standardised screening protocols, implement consistent risk stratification, and provide targeted training for healthcare workers. Embedding these practices within primary healthcare settings could allow for the timely identification of high-risk individuals and facilitate early intervention.

The study demonstrates that strengthening diabetic foot screening at the primary healthcare level is both practical and urgent. It offers a scalable opportunity to reduce the burden of diabetic foot disease, improve patient outcomes, and promote equity in chronic disease management. By incorporating these practices into routine care, South Africa can move towards a more proactive, sustainable, and patient-centred model of diabetes foot care.